

# Using trace elements to identify the geographic origin of migratory bats

Jamin G. Wieringa[1,2], Juliet Nagel[3], David M. Nelson[3], Bryan C. Carstens[1] and H. Lisle Gibbs[1,2]

[1] Department of Evolution, Ecology, and Organismal Biology, The Ohio State University, Columbus, OH, United States of America
[2] Ohio Biodiversity Conservation Partnership, Columbus, OH, United States of America
[3] University of Maryland Center for Environmental Science, Appalachian Lab, Frostburg, MD, United States of America

Corresponding author
Jamin G. Wieringa,
wieringa.3@osu.edu

## ABSTRACT

The expansion of the wind energy industry has had benefits in terms of increased renewable energy production but has also led to increased mortality of migratory bats due to interactions with wind turbines. A key question that could guide bat-related management activities is identifying the geographic origin of bats killed at wind-energy facilities. Generating this information requires developing new methods for identifying the geographic sources of individual bats. Here we explore the viability of assigning geographic origin using trace element analyses of fur to infer the summer molting location of eastern red bats (*Lasiurus borealis*). Our approach is based on the idea that the concentration of trace elements in bat fur is related through the food chain to the amount of trace elements present in the soil, which varies across large geographic scales. Specifically, we used inductively coupled plasma–mass spectrometry to determine the concentration of fourteen trace elements in fur of 126 known-origin eastern red bats to generate a basemap for assignment throughout the range of this species in eastern North America. We then compared this map to publicly available soil trace element concentrations for the U.S. and Canada, used a probabilistic framework to generate likelihood-of-origin maps for each bat, and assessed how well trace element profiles predicted the origins of these individuals. Overall, our results suggest that trace elements allow successful assignment of individual bats 80% of the time while reducing probable locations in half. Our study supports the use of trace elements to identify the geographic origin of eastern red and perhaps other migratory bats, particularly when combined with data from other biomarkers such as genetic and stable isotope data.

## INTRODUCTION

It is estimated that across North America, up to 70 bats per turbine are killed each year (*Cryan, 2008*), totaling ∼600,000 bat casualties annually (*Hayes, 2013*). Based on the number of fatalities of hoary bats (*Lasiurus cinereus*) occurring at wind farms, *Frick et al. (2017)* projected that, in 50 years, the population size of this species may decline by up to 90%. Other results (e.g., *Rodhouse et al., 2019*) support this prediction, highlighting the

threat that the expansion in renewable wind energy represents to bats (*Kunz et al., 2007a*). The majority of the bats killed at wind farms in North America are from three migratory species (eastern red bat, *Lasiurus borealis*; hoary bat, *L. cinereus*; and silver-haired bat, *Lasionycteris noctivagans*: *Kunz et al., 2007a*). Given that the vast majority of bats killed at wind farms are from migratory species, it has been hypothesized that migratory behavior per se may contribute to their susceptibility due to the fact that they encounter multiple wind facilities during seasonal movements (*Cryan & Barclay, 2009*). In support of this idea, bat mortality in North America at turbines has a strong seasonal component, with most deaths occurring during the late summer-fall migration period (*Arnett et al., 2008*).

One consequence of turbine impacts on migratory bats is that it raises questions about the geographic scale over which bat populations could be affected. Specifically, are impacts primarily on local summering populations or, as a result of migratory behavior, more wide-ranging (*Voigt et al., 2012*)? To answer this question, an essential piece of information that is needed is the geographic origin of the individuals killed at specific wind farms. This can then be used to address two important questions: first, as stated above, are bats killed at specific sites from local summering populations or migrants and, second, if they are migrants where did they originate in terms of their summer locations? Such information may inform mitigation efforts and/or decisions about the siting of future wind farms (*Kunz et al., 2007b*).

Accomplishing these goals requires methods for sourcing individual bats to particular geographic locations. At present, few methods exist to generate this information. Techniques such as banding, GPS loggers and transmitters, and large-scale citizen science initiatives have been successfully used to source other migratory animals (*Hobson et al., 2019a*), yet are largely inapplicable to North American migratory bats currently. For example, bats are rarely banded as this activity was ended in the 1970s (*Ellison, 2008*) and now is only commonly used for species that are endangered or threatened (e.g., *Ohio Bat Conservation Plan, 2018*). GPS loggers require recapture of individuals, which is uncommon in many of these species (*Schorr, Ellison & Lukacs, 2014*), and most transmitters are too heavy for use in these small flying organisms (*Kays et al., 2015*). Lastly, due to the nocturnal behavior of bats, citizen science initiatives similar to eBird (*Sullivan et al., 2009*) are limited in scope.

One class of potentially useful markers that have been underutilized to date are intrinsic biomarkers such as stable isotopes, genetic loci and trace elements. Biomarkers offer advantages such as absence of bias to site of marking (i.e., biased due to banding at one location), relatively low cost, a lack of impact on behavior, and small sample-size requirements (*Hobson et al., 2019a*; *Vander Zanden et al., 2018*). However, relative to the use of such approaches to document migratory behavior of birds (*Norris & Marra, 2007*; *Irwin, Irwin & Smith, 2011*; *Hobson et al., 2014*; *Nelson et al., 2015*), the use of biomarkers to track migration in migratory bats is relatively unexplored. To date, the most common type of biomarker used in bats has been stable hydrogen isotopes, which has provided some insights to their migratory behaviors (e.g., *Cryan et al., 2004*; *Fraser et al., 2012*; *Ossa et al., 2012*; *Fraser, Brooks & Longstaffe, 2017*; *Lehnert et al., 2018*). However, hydrogen isotopes are most useful for identifying broad patterns of latitudinal or elevational migration,

whereas they are less useful along longitudinal gradients (*Voigt & Lehnert, 2019*). As a result, other markers are needed, either as standalone biomarkers or for use in conjunction with isotopes (*Hobson et al., 2019b*).

Trace elements represent a class of biomarkers successfully used in birds and other species (*Poesel et al., 2008*; *Szép et al., 2009*; *Ethier, Kyle & Nocera, 2014*; *Flache et al., 2015a*; *Tigar & Hursthouse, 2016*) and so would seem to have potential for sourcing bat movements. Due to differences in the geological history, spatial variation in the concentrations of elements exists across the continental US and Canada (*Smith et al., 2005*). The amount of an element incorporated into tissue of an organism partly depends on the amount of the element present in the soil in the area as each element travels through the food web (*Ethier et al., 2013*). Individuals who forage in the same geographic area are thus expected to exhibit greater similarities in their elemental profiles (*Orłowski et al., 2016*). This type of geographic pattern has been documented in other animals, including humans (*Oyoo-Okoth et al., 2012*), birds (*Burger et al., 2001*), non-human mammals (*Ethier, Kyle & Nocera, 2014*), and insects (*Tigar & Hursthouse, 2016*). This pattern has also been demonstrated in some species of non-migratory bats in relation to the concentrations of mercury (*Chételat et al., 2018*), although *Hernout et al. (2016)* did not find support for this idea in some species of bats for a limited number of elements. Since migratory bats undergo a yearly molt during the summer (*Cryan et al., 2004*; *Britzke et al., 2009*; *Sullivan et al., 2012*; *Fraser et al., 2012*; *Fraser, Longstaffe & Fenton, 2013*; *Pylant, Nelson & Keller, 2014*; *Flache et al., 2015b*), it seems likely that trace elements in their fur should reflect their summering locations.

Here, we investigated the possibility of using trace elements to determine the geographic source of individual bats. Our goals were to: (1) Determine the feasibility of using trace elements to source eastern red bats (*L. borealis*) from different locations from across their range to their summering location; and (2) Provide a basemap and methods for studies seeking to use this methodology. We then compare the precision and accuracy of trace element assignment to other previously studied biomarkers. Our aim is to identify and validate a new class of biomarkers that can be helpful in assessing the geographic scale across which wind farms may impact bat populations.

## METHODS

### Soil data collection

To generate a trace element basemap, data on trace element composition of soil were collected from the US Geological Survey (USGS) and Natural Resources Canada (NRC) online databases available at https://mrdata.usgs.gov/soilgeochemistry/#/summary and https://www.nrcan.gc.ca/home, respectively. We used data that had originally been collected as part of a continent-wide low-density geochemical cooperative survey between USGS, NRC, and Mexican Geological Survey (*Smith et al., 2005*; *Smith et al., 2014*) since previous studies have found little difference between sampling at low-density and high-density on soil trace element interpolation (*Birke, Rauch & Stummeyer, 2015*). However, we were only able to access data for US and Canada. Details of the sampling techniques are found in *Smith et al. (2014)*. In brief, 4857 low density sites (1 site per 1,600 km$^2$) were selected and

samples collected from soil layers of 0–5 cm, Soil A (topsoil), and Soil B+C composite (deeper soil up to 1m). We used trace element concentration from Soil B+C composite samples to minimize the effects of potential surface contaminants (*Sastre et al., 2001*). Elements selected for analysis were the same as those chosen for analysis in fur samples (see below for inclusion criteria) and included Aluminum (Al), Nickel (Ni), Copper (Cu), Rubidium (Rb), Yttrium (Y), Molybdenum (Mo), Tin (Sn), Barium (Ba), Cesium (Cs), Cerium (Ce), Mercury (Hg), Magnesium (Mg), Manganese (Mn), and Iron (Fe).

## Fur analysis

To validate the use of trace elements for sourcing individual bats it is necessary to link a resident bat's fur elemental profile to that of a specific geographic region. To accomplish this, we used fur samples from eastern red bats collected from on axillary region of 126 museum specimens from the Smithsonian National Museum of Natural History that had previously been used for a similar purpose in an isotope validation study (*Pylant, Nelson & Keller, 2014*; Fig. 1). Permission to use these samples in this study was granted by the Smithsonian (J Wieringa, pers. comm., 2019). Following guidelines from isotopic studies (*Cryan, Stricker & Wunder, 2014*), we assume these samples are representative of the element profiles of local bats because these samples were collected during the summer, presumably after molting and prior to fall migration (*Fraser, Longstaffe & Fenton, 2013*). We assume that the fur collected was grown at or near the sampling location. To sample each individual, fur sample was clipped from the torso of each bat and placed inside 2 ml centrifuge tubes for storage until later use.

To prepare samples for trace element analyses, individual fur samples were processed following methods in *Flache et al. (2015a)* and *Hickey et al. (2001)*. Specifically, each sample was cleaned with Triton-X (1:400) and acetone, rinsed with reagent grade water and dried at 60 °C in a sterile Low Density-Polyethylene (LDPE) tube. Gold chloride ($AuCl_3$) was then added at 1 ppm final concentration as a mercury stabilizer (*United States Enivronmental Protection Agency, 2003*). We then added 1 ml of $HNO_3$ and placed tubes on a heating block at 70 °C for 1 h to ensure complete fur digestion. Samples were then allowed to cool to room temperature, after which 10 ppb final concentration Indium was added as an internal standard, and samples were then diluted with reagent grade water to 10 ml.

To more accurately determine trace element concentrations, we used a narrow range calibration curve (calibrations limited to only expected range) as determined from concentrations of previous analyses (Table S1). To create our calibration curve, 5 bat fur samples were analyzed using ThermoFinnigan Element 2 High Resolution ICP-MS at the Trace Element Research Lab (TERL) at Ohio State using the Semi-Quantitative mode (*Chen et al., 2008*; Table S1). This mode allows for rapid quantification of all elements through estimating only a few elements accurately and inferring the rest on previously determined ratios. While there is increased error, this method of analysis allows for rapid quantification to assess which elements to use in further analysis. When compared to blanks, these 5 samples identify elements that were reliably detected above background and gave concentrations that were used to create narrow range calibration standards. This more rapid analysis allowed us to better determine which concentrations to expect for
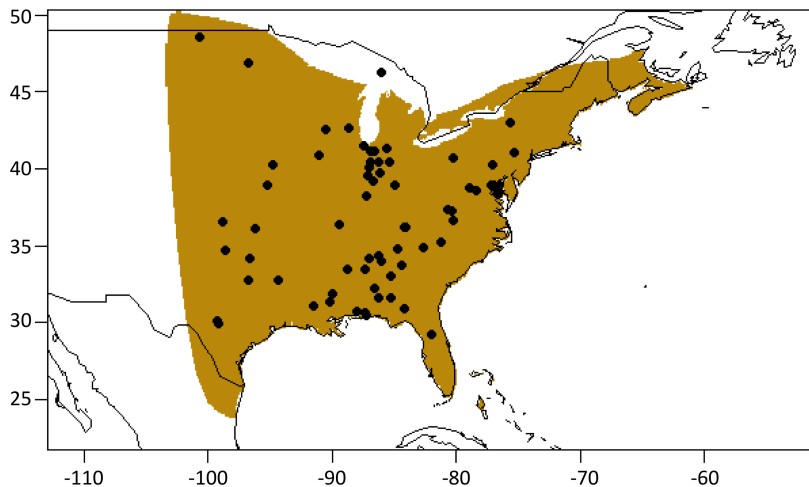

**Figure 1** **Sampling distribution for _L. borealis_.** Distribution of eastern red bat samples used for validation. Samples were obtained from Smithsonian Museum of Natural History. The brown shaded area represents the current IUCN Red List range (_Arroyo-Cabrales et al., 2016_). Map outline of countries generated using 'maps' R package (_Becker et al., 2018_).

more in-depth determination (below) which created more accurate calibration curves, and allows for better determination of concentrations.

We used the normal quantification mode on the ThermoFinnigan Element 2 ICP-MS to make additional elemental concentration measurements. Samples were fed into the ICP-MS with a delay before reading to reduce the chance of carry over effects, followed by a short wash between each sample. Samples were then ionized in plasma and the concentrations determined when compared to a calibration curve. After every 10-15 samples, one of the calibration standards used to create the calibration curve and a blank were re-analyzed to ensure accuracy of our results. Additionally, ten samples were re-run to determine analytical variation. We used software included with the ICP-MS to correct each sample to the values for the internal standard and further corrected sample concentrations based on the mass of the fur used.

## Statistical analyses

Key steps in our statistical analyses were as follows: (1) Summarize trace element variation among individual bat fur samples using spatially explicate measures; (2) Link fur and soil elemental concentrations across the range of this species; (3) Generate a basemap across the species range; (4) Assign individuals to origin, and assess the accuracy and precision of assignment.

## Summarizing individual variation

Sourcing individuals via their trace element profiles requires summarizing the spatial correlation of trace elements among individuals sampled (_Donovan et al., 2006_). Therefore, we used sPCA scores to summarize variation for bat fur concentrations using the function 'gwpca' in the 'GWmodel' package in R (_Gollini et al., 2013_). sPCA introduces a spatially
based weighting matrix into the PCA matrix. The advantage of sPCA over traditional PCA is that sPCA maximizes spatial autocorrelation, and therefore shows strong spatial relationships in the first few principle axes. When using 'gwpca' we generally applied the default values in the package, with the exception that bandwidth was determined using the 'bw.gwpca' function. This function plays an important role in the weighting function of the sPCA. The loadings were then used to determine the principle component scores based on element profiles for individual bats. These values could then be used for geographic assignment; both for creating a basemap, and validating their use.

## Linking fur and soil trace element values

We first tested the assumption that element values from fur reflect values in the soil from the same geographic location. To do this we generated linear models for individual and multivariate measures of element abundance which estimated the relationship between soil element concentration (extracted from maps in *Smith et al., 2014*) and fur element concentration. This was accomplished using the function 'lm' in R (*R Core Team, 2019*). The usefulness in assignment was assessed by the amount of variation explained by the linear model ($R^2$) and the significance of the relationship. However, we found weak relationships for individual elements, which precludes the use of single elements for assignment. Weak univariate relationships does not, necessarily, preclude significant multivariate relationships from being present. In which case, principle components generated above may capture patterns of variation across multiple elements that are useful for assignment. To assess the ability of use of a composite metric (principle components), we compared predicted and actual values of PCs, and compared soil to hypothetical fur to PCs (described below) to actual fur PC values.

## Geographic assignment

We used methods developed for sourcing species using hydrogen isotope values of fur (e.g., *Pylant et al., 2016*) to conduct geographic assignment analyses. We created a basemap of multivariate estimates of fur values in ArcMap 10.6, after removing outliers using squared robust Mahalanobis distances and evaluating plotted values (*Ethier, Kyle & Nocera, 2014*). When creating our base map, we used two methods to generate two distinct portions of the base map (Fig. S1). The first portion was restricted to the geographic range of our samples (Fig. 1; ∼92% of range) and was determined using the first principle component value determined solely from the fur concentrations of the 14 elements from our validation data set based on 111 samples. The other 15 samples were used for independent verification of the method (below). *Wunder & Norris (2019)* characterized this type of model as one that directly links tissue chemistry and geography. Fur principle component values were taken from the sPCA analyses, with values interpolated between the locations of our bats using inverse distance weighting (IDW). However, because this approach only uses fur concentrations from samples collected, we are only able project to the limit of our points on a map which does not encompass the entire range of the species (Fig. 1).

To extend our prediction coverage to the second portion (∼8% of range; Fig. 1) of our base map for which no bat samples were available we related soil element concentration

to the fur concentrations within the first portion that covered all sampled points. Using those relationships, we transformed soil concentration outside of the sampling extent to hypothetical fur concentrations (Fig. S1). Using the sPCA loadings determined previously, we then transformed the hypothetical fur to PC1 values. *Wunder & Norris (2019)* describe this approach as an indirectly linked tissue chemistry and geography method. This method is less accurate as more noise is present in the samples due to elemental concentrations being filtered through the food web from soil to fur. As the two maps (directly linked and indirectly linked tissue and geography) do not overlap at any point, they were then merged to create one continuous joint projection across the US and Canada. There was a slight misalignment between the two portions which was left in place to preserve the more accurate direct fur-to-fur comparison present in the base maps generated. Additionally, we tested all metrics for assignment (below) using both the merged maps and the direct geographic relationship (portion 1) separately to ensure adding the second portion did not impact our results.

When generating our base maps we used a leave-three-out (n-3) approach for cross validation. For each process in creating the base map, n-3 samples were used, leaving 3 samples to be assigned back to the base map and testing the performance of our model. This was repeated 100 times, allowing us to assess how our analysis will translate to an independent dataset. When assigning individual bats back to their origin, we used methods originally used in assigning migratory species using hydrogen isotopes values present in bird feathers and bat fur (*Fraser et al., 2012*; *Tonra, Both & Marra, 2014*; *Fraser, Brooks & Longstaffe, 2017*). We did this using the normal probability density function as given in the following equation (*Tonra, Both & Marra, 2014*):

$$f(y^*|\mu_i,\sigma) = \frac{1}{\sigma\sqrt{2\pi}} e^{\frac{-(y^*-\mu_i)^2}{2\sigma^2}}$$

where $f(y^*|\mu_i, \sigma)$ is the likelihood that an individual with a given principle component fur value ($y^*$) originated from cell $i$, $\mu_i$ is the predicted sPCA value of cell $i$, and $\sigma^2$ is the variance ($\sigma$ equaling the standard deviation) within a sampling area which was set at the standard deviation of residuals from the regression between predicted and actual values ($\sigma^2 = 9.696$). The result of this equation is the probability, given a measured fur PC value, came from the cell in question, ranging from 0 to 1. We then iteratively analyzed every cell ($0.16° \times 0.16°$; ~315 km$^2$) of each base map for all individuals and determined the probability a given individual originated from all cells present on the base map.

To evaluate the quality of our model we estimated the probability of assignment for the location where the sample had been collected and considered a prediction ≥50% as correct. We chose 50% as our cut off because we prioritized accuracy (i.e., a correct prediction) above precision (i.e., reducing area of prediction). However, we also evaluated cut-offs of 66% and 75% accuracy as have been used previously (*Tonra, Both & Marra, 2014*; *Seifert et al., 2018*). Finally, due to the volant nature of these species, if the prediction value at the sampling location was below 50%, we checked if a point within 10 km (potential nightly travel distance from roost; *Hayes & Wiles, 2013*) was above our threshold. If a point

within 10km was above our 50% threshold, we also considered this a correct assignment to account for the possibility of nightly bat movement around the capture location.

### Independent fur sample validation

For further validation we used a subset of random samples from the original 126 ($n = 15$) that had not been used in base map generation or the leave-three-out analysis. Our logic was that if samples that are independent from the rest of the analyses can also be accurately sourced to their inferred location of origin, it supports the claim that this method can be applied to all *L. borealis* and can be used in future analyses. Each sample's fur trace element concentrations were transformed given the sPCA component values previously determined and assigned to a geographic area following the same protocol above.

## RESULTS

### Trace element concentrations

Concentrations of trace elements in the fur of eastern red bats are similar to values previously reported in fur of other bats (Table 1; for detailed summary see Table S2; *Hickey et al., 2001*; *Zocche et al., 2010*; *Flache et al., 2015a*; *Flache et al., 2015b*); in addition, levels of variation (reported as SD of mean concentrations) were also similar. Overall, there is a high level of variation in trace element concentrations present among individual samples with all elements having a standard deviation greater than 50% of the mean (Table 1). This suggests that values for individuals collected from different locations differ greatly, which increases the likelihood of correctly assigning individual bats to source areas.

### Linking fur and soil element values

All individual elements showed only weak correlations between element values in fur and in the soil with all $R^2$ values $< 0.16$ and only three showing a statistically significant relationship ($P < 0.05$) (Table 1). This means that individual elements are unlikely to provide sufficient resolution for sourcing individual bats.

Next, we explored the use of composite measure that used multivariate approaches to combine data from different elements in fur. We used a spatial principle component analysis (sPCA) approach to reduce the number of variables from 14 elements to a set of spatial weighted principle component values. We first used Kaiser criterion to evaluate our principle components (PC) from our sPCA from 'gwpca' in R (*Gollini et al., 2013*), with the recommendation to drop PCs with an eigen value below 1.0. By only retaining PCs with eigenvalues above 1.0, we retain factors that extract at least as much information as an equivalent single variable (*Kaiser, 1960*). In addition, we examined the percent of variation explained by each principle component value. PC 1 explained 50% of the variation present in our data with PC 2 only adding 12% variation explained (Table 2). PCs 3 and larger also added very little to the cumulative variation explained (Table 2). As a consequence, we chose to only retain PC 1 for further analyses. Examination of the first PC shows Al, Cu, Ba, Mg, and Fe had the largest absolute weights (PCA loadings multiplied by average fur element values) on PC 1 relative to their means with values greater than or equal to 1.

The Monte-Carlo permutation test of the sPCA score was not significant ($p = 0.07$), likely due to similar elemental concentrations in areas that are geographically separated

**Table 1  Mean concentrations and deviations for trace elements.** Mean concentrations of trace elements in *L. borealis* fur from across their range with the standard deviation (SD) also reported. For each element the correlation between fur and soil (using $R^2$) concentrations are presented with the associated $p$-value. Those that are significant ($p < 0.05$) are designated with an '*'.

|  | Mean | STD | Soil/Fur $R^2$ | Soil/Fur $p$-value |
|---|---|---|---|---|
| Al (ppb) | 18.814 | 15.537 | 0.010 | 0.21 |
| Ni (ppb) | 2.254 | 2.663 | 0.010 | 0.45 |
| Cu (ppb) | 6.237 | 3.053 | 0.015 | 0.35 |
| Rb (ppb) | 0.652 | 0.352 | 0.073 | 0.04* |
| Y (ppb) | 0.378 | 0.199 | 0.062 | 0.06 |
| Mo (ppb) | 1.585 | 0.805 | 0.154 | 0.01* |
| Sn (ppb) | 4.225 | 5.169 | 0.071 | 0.04* |
| Ba (ppb) | 5.415 | 3.010 | 0.001 | 0.77 |
| Cs (ppb) | 0.914 | 2.609 | 0.002 | 0.34 |
| Ce (ppb) | 0.325 | 0.159 | 0.001 | 0.62 |
| Hg (ppb) | 4.777 | 4.431 | 0.004 | 0.63 |
| Mg (ppb) | 51.891 | 24.056 | 0.120 | 0.46 |
| Mn (ppb) | 1.297 | 0.677 | 0.021 | 0.27 |
| Fe (ppb) | 65.486 | 80.390 | 0.001 | 0.78 |

**Table 2  sPCA loadings and weights for each element.** sPCA loadings and weights (loading*average element value) for each element onto the first three principle components, as determined using 'gwpca' in R. The proportion of the variance, or the amount of variation explained by each principle component, is at the bottom of the table.

| Element PC1 | Loading PC1 | Weight PC2 | Loading PC2 | Weight PC3 | Loading PC3 | Weight |
|---|---|---|---|---|---|---|
| Al | 0.07 | 1.39 | 0.15 | 2.77 | 0.39 | 7.41 |
| Ni | 0.23 | 0.51 | −0.06 | −0.12 | −0.17 | −0.38 |
| Cu | 0.21 | 1.28 | 0.06 | 0.35 | 0.21 | 1.32 |
| Rb | 0.27 | 0.18 | −0.05 | −0.03 | −0.18 | −0.12 |
| Y | 0.27 | 0.10 | −0.06 | −0.02 | −0.19 | −0.07 |
| Mo | 0.27 | 0.43 | −0.04 | −0.06 | −0.19 | −0.30 |
| Sn | 0.15 | 0.61 | −0.03 | −0.13 | 0.09 | 0.38 |
| Ba | 0.28 | 1.51 | 0.02 | 0.12 | 0.04 | 0.23 |
| Cs | −0.03 | −0.03 | 0.02 | 0.02 | 0.08 | 0.08 |
| Ce | 0.27 | 0.09 | −0.06 | −0.02 | −0.16 | −0.05 |
| Hg | 0.16 | 0.78 | 0.06 | 0.26 | 0.30 | 1.45 |
| Mg | 0.18 | 9.55 | −0.01 | −0.52 | 0.03 | 1.71 |
| Mn | 0.06 | 0.08 | 0.53 | 0.69 | −0.18 | −0.23 |
| Fe | −0.02 | −1.18 | 0.56 | 36.61 | −0.15 | −10.08 |
| Proportion of variance | | 0.50 | | 0.12 | | 0.10 |

from each other. This indicates that global spatial structures exhibited by elements are not likely due to random variation. To determine if the PC1 values offer a better way to assign individual bats than single elements we compared predicted to actual PC values using a linear model. The PC1-based model had an $R^2 = 0.5$ ($p < 0.001$) which is much

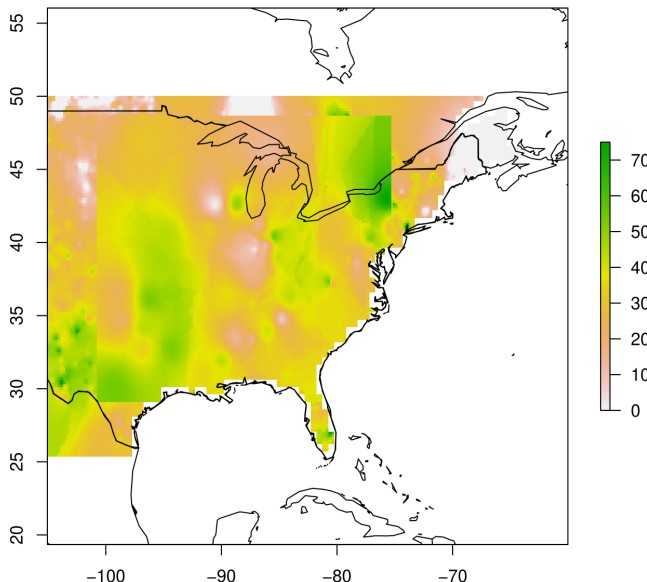

**Figure 2** **Basemap generated from sPCA results for PC1.** Colors represent component scores, to which individuals can be assigned based on the component scores for their fur. Map outline of countries generated using 'maps' R package (*Becker et al., 2018*).

higher than the values for individual elements. Consequently, we used PC1 values as our measurement of element variation in subsequent analyses.

## Geographic assignment using trace elements

To assess our ability to conduct the geographic assignment of individual bats, we explored three distinct but related factors: accuracy, precision, and geographic patterns. Accuracy is defined by the ability to correctly assign an individual to their origin. Precision is assessed by the amount of area determined as being probable as an area of origin (less area predicted equals higher precision). Finally, geographic patterns are patterns that follow an ordered arrangement through space (e.g., a north/south gradient).

To explore these aspects of our ability to do assignments for the independent dataset, after generating our leave-three-out assignment, we combined the averaged basemaps into one single basemap surface (Fig. 2). Extending our base maps by relating soil to hypothetical fur to principle component values showed a correlation of 0.24 between direct and indirect base map generation. Further, when creating our base maps we compared the merged raster and direct geographic assignment; both showed similar results (accuracy remained consistent, precision ±1%) and so we used the merged basemap for assignment.

### *Accuracy*

We measured accuracy as the number of individuals with a probability over 0.5 or being within 10 km of an area above 0.5. Other threshold values (0.66 and 0.75) that were tested resulted in poor assignment accuracy (56% and 49% correct) and so were not considered further. The probability for the training dataset to be accurate at a 50% threshold was 81%, and was determined by the number of individuals above 0.5 using a leave-three-out

**Table 3  Locations and results for 15 independent samples.** Locations and prediction information for 15 test individuals and the training dataset following a leave-three-out approach. We report the PC values for each individual, the prediction probability at the sampling location, and the percent of *L. borealis* range predicted.

| Longitude | Latitude | PC values | Probabilty | Range percent |
|---|---|---|---|---|
| **Training** | **Dataset** | **Avg** | **0.81** | **0.49** |
| −86.10 | 37.19 | 20.43 | 0.52 | 0.41 |
| −77.04 | 38.91 | 34.10 | 0.94 | 0.76 |
| −82.60 | 38.41 | 97.99 | 0.00 | 0.01[b] |
| −96.18 | 36.10 | 52.52 | 1.00 | 0.21 |
| −84.23 | 30.91 | 38.45 | 0.99 | 0.72 |
| −76.87 | 38.78 | 28.79 | 0.80 | 0.71 |
| −82.60 | 34.92 | 23.99 | 0.34[a] | 0.71 |
| −97.04 | 32.90 | 15.29 | 0.74 | 0.22 |
| −77.14 | 38.97 | 26.38 | 0.51 | 0.65 |
| −95.23 | 38.97 | 42.70 | 0.95 | 0.59 |
| −99.11 | 29.89 | 50.33 | 0.95 | 0.29 |
| −84.44 | 33.71 | 92.99 | 0.00 | 0.03[b] |
| −75.48 | 41.39 | 17.40 | 0.32[a] | 0.44 |
| −96.76 | 46.88 | 31.58 | 1.00 | 0.75 |
| −90.56 | 42.59 | 41.69 | 0.01 | 0.62[b] |

**Notes.**
[a] Denotes values below 50% threshold, but within 10 km of possible sources (above 50%), and are considered successful.
[b] Denotes those that were incorrect and therefore not included for deteremining precision.

analysis (Table 3). Additionally, 12 of our 15 validation independent samples were either above 50% or within 10 km of a cell above our threshold, yielding an accuracy of 80% (Table 3). Because of bats being able to fly ∼10 km from their roost in a given night to feed, we considered these correct assignments. Therefore, using a threshold of 50%, we correctly predict the sampling location ∼80% of the time. Examples of a correct and incorrect assignment from our cross-validation dataset are shown in Fig. 3 (all assignment maps from cross validation in Fig. S3; same order as Table 3).

### Precision

Precision for geographic assignment, or area predicted above 0.5 threshold compared to total area, is given in the Range Percent column in Table 3. Of note, if an individual was not correctly assigned they were not included in calculations of precision. When using the samples from the leave-three-out testing and map generation, we on average, reduced the possible range by 51%. When using our 15 independent validation samples, for individuals that were predicted correctly, our average prediction range is 47% (minimum: 22%, maximum: 76%) of the total possible range these bats could have originated from. To summarize, when we have a bat from an "unknown" origin, we can reduce on average the possible source locations for this individual by over 50%.
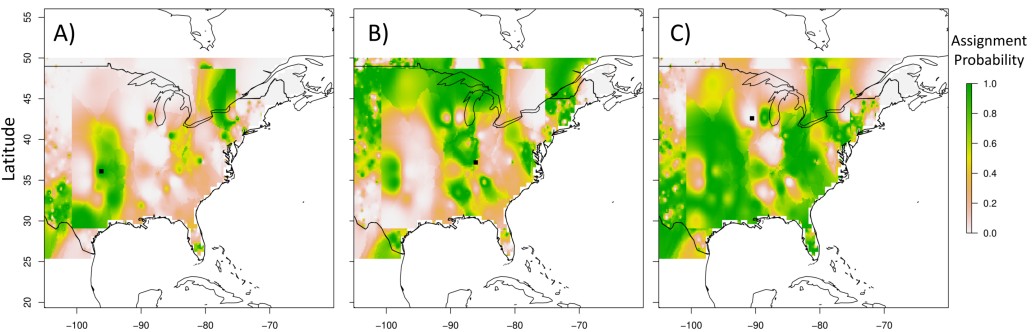

**Figure 3** **Sample prediction surfaces generated.** Prediction surfaces (scale bars ranging from 0% probability to 100% probability) for three different bats. Black dot represents sampling location. A and B correspond to correct assignment, while C is an incorrect assignment of origin. Map outline of countries generated using 'maps' R package (*Becker et al., 2018*).

### *Geographic patterns*

Finally, while the precision of sample assignment was lacking for some samples (Table 3), we do observe geographic variation. However, we do not observe a clear geographic pattern present in other aspects of assignment accuracy when overlaying the boundaries of predicted sources (Fig. S4). While this precludes use of trace elements to clearly delineate north/south or east/west movements, when combined with other markers that exhibit these patterns trace elements could add overall precision to assignment accuracy. This is especially true for hydrogen isotope assignment probabilities, as the base map generated here is different from that for hydrogen isotopes based on correlations done in R on values extracted from both maps from 150 random locations ($R^2 = -0.13$, $p = 0.11$).

## DISCUSSION

Our results show the promise of using trace elements to accurately assign individual bats back to their summer molting location. Specifically, we observe a correct assignment rate of ~80% with a reduction in predicted area of ~50% from total range. It is important to consider that there is a trade-off between accuracy and precision in that the number of correct predictions (true positives) is inversely related to the number of incorrect absences (false positives). For example, if we wished to increase the number of correctly predicted origins (accuracy), we would need to increase the number of false positives predicted thereby decreasing our precision (*Pearce & Ferrier, 2000*). In concrete terms, if a bat originates near the edge of a predicted area, by attempting to increase our precision (reduce prediction area) we could lose our ability to make a correct prediction of the bats origin. In this study, we chose to emphasize accuracy because for species with conservation implications it is often times true that accuracy is more important (*Allouche, Tsoar & Kadmon, 2006*). In other words, it is better to know where an individual is than to identify a smaller range that may not be correct.

Our analyses make several key assumptions. The most important is that the fur of a sampled bat was grown at (or very near to) the sampling location, hence the element profile

matches that of a particular location. As noted previously, eastern red bats are known to molt during the summer (*Fraser, Longstaffe & Fenton, 2013*) within a relatively short time frame, and as a result we assume that fur is grown where they spent their summers. As such, it is reasonable to assume that the bats used in this study were captured near the location where they grew new fur. Second, we only used samples that had confirmed geographic origins using isotope analysis (*Pylant, Nelson & Keller, 2014*). Both these features of the samples we used to validate our method make us confident our assumption about the geographic origin of our fur samples is accurate.

Multiple studies have investigated the amount of trace elements present in fur and other tissue of various bat species to track bioaccumulation (*Hickey et al., 2001*; *Flache et al., 2015a*; *Flache et al., 2015b*; *Rahman, Talukdar & Choudhury, 2020*). For example, *Chételat et al. (2018)* showed a correlation between the atmospheric mercury in Canada and the concentration present in the fur of a non-migratory bat, supporting the notion of the use of mercury levels to source bats. We examined the possibility of using atmospheric mercury but found no significant association between levels in fur and geographic location (Fig. S2). More broadly, the use of trace elements to geographically source individuals has been successful in other species such as badgers (*Ethier, Kyle & Nocera, 2014*), tobacco cutworm (*Lin et al., 2019*), and giraffes and elephants (*Hu, Fernandez & Cerling, 2018*). Our work adds to these results by providing an example of using trace elements to source individual bats on a broad geographic scale.

We focused on a single bat species, but our results highlight the possibility of using trace elements to source individuals in other species of migratory bats because of their similar life history features (*Harvey, Altenbach & Best, 2011*). Due to the broader or similar diet in other insectivorous migratory bat species, we suspect that we may find better resolution for other species (*Hickey, Acharya & Pennington, 1996*; *Chételat et al., 2018*) because a broader diet is likely to better reflect the trace element profile of a particular location due to an increased range of element sources from prey (e.g., *Chételat et al., 2018*). We expect that due to the similar diets between eastern red bats and hoary bats (*Newbern & Whidden, 2019*) that this method will be applicable to hoary bats with the same level of assignment accuracy. *Wilson (2017)* furthered showed no significant difference in diet between hoary and eastern red bats, demonstrating that lack of statistically different diets is widespread across the range (Pennsylvania (*Newbern & Whidden, 2019*) and Kansas (*Wilson, 2017*)). *Newbern & Whidden (2019)* also found that silver-haired bats had a different and broader diet than hoary and eastern red bats, indicating that the method may work better for this species due to their broader diet.

In terms of diet stability over time and its potential impacts on element profiles, while dietary preference switches are known to occur in eastern red bats these shifts take place just prior to or during migration (*Hayes et al., 2019a*), indicating a relatively stable diet over a summer molting period hence minimal impacts on element variation as assayed here. Some other species, such as hoary bat, similarly do not appear to shift their diet during summer (*Reimer, Baerwald & Barclay, 2010*) suggesting minimal impacts on element profiles.

Finally, elemental sourcing is likely applicable to bats on other continents where substantial numbers of the individuals killed by wind turbines are migrant species (e.g.,

Europe - *Lehnert et al., 2014*) with similar diets (*Vaughan, 1997*), and an availability of soil data on element abundance (*Birke, Rauch & Stummeyer, 2015*). However, we also note that because the specific characteristics of elemental uptake can vary between even closely related species, the use of elements for sourcing individuals needs to be validated for each species (*Wunder & Norris, 2019*), as is similar to other biomarkers.

Two other classes of biomarkers that have been evaluated for use in sourcing bats are genetic markers and stable isotopes. To date, three studies have attempted to determine the feasibility of using genetic markers by looking for genetic structuring (*Vonhoff & Russell, 2015*; *Pylant et al., 2016*; *Sovic, Carstens & Gibbs, 2016*). However, none of these studies have found structuring for the three species of migratory bats in North America. Currently, studies are underway to determine the feasibility of using other types of genetic markers for sourcing bat include whole genome sequencing (J Wieringa et al., in prep.) and genotyping by sequencing (GBS) (J Nagel et al., in prep.).

In contrast, stable isotopes (mainly hydrogen) have shown more promise for determining origins of latitudinal migratory bats (e.g., *Baerwald, Patterson & Barclay, 2014*; *Pylant et al., 2016*). Depending on the region, accuracy of assignment for isotopes can range from ∼50% to 90% with reasonable precision (*Popa-Lisseanu et al., 2012*), which is comparable to the accuracy of the trace elements used here, although isotopes have higher precision than trace elements (∼33%–50% vs 50% of range, respectively; *Ruegg et al., 2017*). Both trace elements and isotopes require the same type and roughly same amount of sample, but trace elements cost slightly higher (∼$15 vs $25 per sample; J Wieringa, pers. comm., 2018). Finally, an important difference in the information that each marker type yields is that trace elements provide geographical sourcing along both longitudinal and latitudinal gradients, depending on region, whereas isotopes provide discrimination primarily along latitudinal gradients (*Hobson et al., 2019b*).

Combining data from multiple markers or methods of analysis may be the most productive way to identify the geographic origins on individual bats as has been shown to be the case for birds (*Hobson et al., 2019b*). For example, in other bat species the use of multiple markers enabled levels of accuracy to be maintained while reducing the area of residence predicted for individuals (*Popa-Lisseanu et al., 2012*). In some bird species, combining isotopes with trace elements has been shown to be more effective in sourcing individuals to specific locations than using data from just one type of marker (*Gómez-Díaz & González-Solís, 2007*). Finally generating occupancy prior surfaces based on Species Distribution Models (SDMs) may also be useful as not all locations are equally likely to be a source for a given individual due to environmental differences across space (*Cryan, Stricker & Wunder, 2014*). Statistical approaches, such as cross-validation calibrated combined model tuning, in which probability surfaces based on SDMs, and data from trace elements, isotopes, and possibly genetics are combined offer a way to incorporate distinct information from different markers to generate a single assignment probability for a given sample (*Rundel et al., 2013*).

We emphasize that using approaches such as trace element analyses to mitigate wind turbine impacts on bats has economic as well as population-related benefits. Across the United States, all bats, including eastern red bat, contribute >$3.7 billion annually to the

economy through the consumption of pest insects (*Boyles et al., 2011*). Multiple studies have found economically important pest species in the stomachs of eastern red bats (e.g., *Clare et al., 2009*), including gypsy months, tent caterpillars, and cutworms. Tools such a trace elements can aid in mitigating these projected declines of migratory bat populations due to wind energy impacts and preserving these benefits (*Frick et al., 2017*).

## CONCLUSION

To advance migratory bat conservation, it is vital that we understand their migration patterns (*Kunz et al., 2007b*). Identifying patterns of movement would allow planning the locations of wind farms to mitigate impacts on bat populations (*Cryan, Stricker & Wunder, 2014*). This information could also be used to set standards for "smart" curtailment (*Hayes et al., 2019b*) to reduce mortality and better plan future developments in wind energy that minimize placement in movement corridors. Toward this goal, we find that trace elements are a viable method for sourcing eastern red bat and this approach may be applicable to other migratory bat species. The grand challenge in understanding migration is to know when, where, why, and how animals migrate (*Bowlin et al., 2010*). Future studies need to explore not only the singular use of trace elements as a sourcing method, but also the process of combining trace elements with multiple potential sources of geographic information, such as isotopes, species distribution models, and genetics to accomplish this goal.

## ACKNOWLEDGEMENTS

We thank the Trace Element Research Lab (TERL) at The Ohio State University for their assistance with the processing of samples. Specifically, we thank Anthony Lutton for help with scheduling lab equipment use and for help with interpreting our results and John Olesik and other members of the TERL for general assistance. We also thank the Carstens and Gibbs labs in the Department of EEOB at Ohio State for their advice and assistance with the editing of this manuscript. Finally, we thank Erin Hazelton and Jonathan Sorg, Ohio Division of Wildife, for assistance with grant administration. This study is a contribution from the Ohio Biodiversity Conservation Partnership. Samples for this project were provided by the Smithsonian Institution, National Museum of Natural History, Division of Mammals.

### Funding

This work was supported by a grant (GRT00046616) from the Competitive State Wildlife Grants Program to Ohio State University and the University of Maryland Center for Environmental Science jointly administered by the US Fish and Wildlife Service, the Ohio Division of Wildlife and the Maryland Division of Natural Resources. The funders had no role in study design, data collection and analysis, decision to publish, or preparation of the manuscript.

## Grant Disclosures

The following grant information was disclosed by the authors:
Competitive State Wildlife Grants Program Ohio State University: GRT00046616.
University of Maryland Center for Environmental Science.
US Fish and Wildlife Service, the Ohio Division of Wildlife and the Maryland Division of Natural Resources.

## Competing Interests

David M. Nelson is an Academic Editor for PeerJ.

## Author Contributions

- Jamin G. Wieringa conceived and designed the experiments, performed the experiments, analyzed the data, prepared figures and/or tables, authored or reviewed drafts of the paper, and approved the final draft.
- Juliet Nagel, David M. Nelson, Bryan C. Carstens and H. Lisle Gibbs conceived and designed the experiments, performed the experiments, analyzed the data, authored or reviewed drafts of the paper, and approved the final draft.

## Data Availability

Raw concentrations are available in the Supplemental Files and all data and code are available at Github: https://github.com/jgwieringa/Trace_Origin.

## Supplemental Information

Supplemental information for this article can be found online at http://dx.doi.org/10.7717/peerj.10082#supplemental-information.

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
