# Peer review of "Using trace elements to identify the geographic origin of migratory bats"

_PeerJ, doi:10.7717/peerj.10082_

## Round 0.1 · original submission · Minor Revisions

· Academic Editor

Minor Revisions

As you can see, all reviewers see merit in your work and find it worthy of publication. All reviewers also suggest improvements to your manuscript. Only one of the reviewers suggests extended re-analyses of the specimens. I am always hesitant to require new experiments, but of course, there are cases when these are warranted. Thus, please make abundantly clear, in case you do not concur with this reviewer, what your reasons are to not conduct the suggested analyses.

Reviewer 1 ·

Basic reporting

The goal of this work was to use the trace element composition of bat fur to infer geographic origins. The paper is well written, logically organized, and should attract interest. That said, I’m concerned about a number of issues. First, the scale of the geochemical data. As the authors point out, the original surveys were very coarse (one site per 1600 km2) and this seems to present a major mismatch with respect to the objectives, especially when precision is interpreted at the 10’s of km scale. Further, the geochemical data surface is very different than the hydrogen isotope isoscape, which has much better data density and interpolation heavily based on known relationships (i.e. orography, elevation, etc.). Such constraints on geochemical data are likely lacking. Second, the training data (fur) seems very limited in both density and space; the actual sample size is never mentioned. Additionally, the relationship between fur and soil is very weak, with the highest R2 explaining about 15% and only three elements yielding significant R2 values (most with very low R2 values). Lastly, I’m concerned that the input functions for elements into fur were all treated similarly and this is likely problematic. In other words, some elements bioaccumulate to a greater extent than others (for example, Ni accumulates more than Hg). And to bring this back to hydrogen isotopes, the input function is used in assignment models. I do note, however, that the authors disclose all of these problems early on in the results section and then adopt a multivariate solution to assess assignment accuracy and precision. Despite the design issues, I do think this is worthy of publication because it highlights the knowledge gaps surrounding the use of elemental data for assignment purposes. My recommendations are to discuss i) the coarse spatial resolution of the geochemical data and how that might impact the efficacy of using such data for assignment purposes (this was not discussed, yet the early results demonstrate the problem in my opinion) and ii) better address the elemental input function issue (Al, Ba, Cu, Fe, Mg – three of the five are essential).

Experimental design

no comment

Validity of the findings

no comment

Reviewer 2 ·

Basic reporting

This paper sought to use trace element analysis results from fur samples taken from bats to assign origins (e.g. location of fur growth) to individual bats. The eventual goal is to use this technique on unknown-origin migrants, but this paper focused on developing and ground-truthing the method using known-origin samples. Results were promising, with the trace element profiles of the fur samples corresponding with those of previously taken soil samples at the location of fur growth. Assuming a probability of origin threshold of 0.5, bats could be assigned to their location of origin about 80% of the time and the total area from which bats may have originated was decreased to about half of the species total range.

The paper was fairly well written with a clear list of goals. There were some unclear parts and a quite a few small typos, which I itemize in the general comments section below. The introduction provided an appropriate and thorough background on the subject matter. The figures and tables were well done, although I do not understand Figure 4. It is not clear to me what it is showing and so I cannot comment on whether it is necessary; the caption needs to be clearer.

In my opinion, the discussion is the weakest part of this otherwise very strong paper. I thought that several parts of it could be removed. For example, the whole paragraph on assumptions about the timing of fur growth really repeated what was in the introduction and didn’t need to be there. Also, the paragraph talking about previous studies on genetics basically concludes that so far, we can’t use genetics at all to assign origins to members of the study species. This could have been summarized in one sentence. I think the section talking about the impact of diet on the trace element composition of tissues and the section talking about the future utility of combining multiple origin assignment techniques are the most important parts of the discussion. There could be more on diet – currently only one paper is cited that talks about the diet of the study species and one other species and this paper is quite old. Much more work has been done on dietary preferences of these species since then. Could the authors elaborate on the findings of this more recent work and talk about specifically which trace element measurements might be impacted by a broader diet? What about in other species?

Experimental design

The research questions are well defined and explicitly stated. The introduction makes clear how the study is contributing to filling a significant knowledge gap (adding to the scant literature about trace elements in bat fur, using these data for origin assignment). To my eye, the statistical approach was extremely rigorous and clearly explained. The paper worked through four separate statistical processes and these seemed logical and according to best practices based on my own experience, which has mainly been working with origin assignment models based on stable hydrogen isotope results.

My one criticism is that I found the methods/results explaining the linkage between fur and soil trace element profiles to be very confusing. The methods state that linear models were used to make this link and in the results, I see that there were no strong relationships between individual trace elements in fur and soil. The results then go on to describe the results of multivariate analyses and I’m a little fuzzy on the details (line 277). I understand that PC1 scores were used to summarize the trace element profiles of the fur samples. Were sPCA analyses also done on the trace element profiles of the soil samples? Were the linear models looking at the relationship between PC1 of fur and PC1 of soil? Or am I misunderstanding? I would appreciate this being explained more clearly.

With respect to ethical considerations, the paper should state that the Smithsonian Museum approved the present use of their samples for trace element analysis (as the paper indicates that fur samples were taken for a previous project in 2014 on stable hydrogen isotopes and museums often approve destructive sampling of specimens for specific projects and require additional explicit permissions if those samples are to be used for other projects).

Validity of the findings

All data are provided, but I have a suggestion: I really like figure 3 in the manuscript and recommend it be retained as it is, however in the spirit of sharing all data (as requested by PeerJ), the authors might consider providing maps like the ones in figure 3 for all individual bats in the study (in a supplemental file). The conclusions are appropriate. Based on my own experience, analyses appear well done, but I do not have experience in trace element analysis and cannot critique the description of the laboratory techniques.

Additional comments

Overall – Personally, I do not like the term “tree-roosting bats” to refer to the trio of bats that are the focus of the introduction (eastern red bats, hoary bats, silver-haired bats). There are many other bat species that roost in trees and in many instances where the focal species have been found roosting out of trees. I realize that “tree roosting” is a fairly common term in the context that the authors are using it, but I’ve seen pushback on this terminology in other places and I tend to agree. I would recommend simply referring to “migratory bats in North America”. I’ve also seen “latitudinal migrants” used (to differentiate from hibernating species who may migrate in all directions), but more recent work looking at the movements of these three species even suggests that “latitudinal” is an oversimplification.
Line 19 – I think “source” should be plural
Line 39 – You might consider citing Rodhouse et al. (2019) here as well, as they come to similar conclusion.
Line 45 – I would clean up the sentence that states that “most bats that die are migratory”. In fact, I don’t think there is a lot of direct evidence that bats that die a wind energy facilities are migratory (cite the few papers that do demonstrate this). The main evidence for the bulk of the mortality being migrants is the timing of the mortality: during fall the migratory period. Be precise in wording: e.g. “Most bat mortality occurs during the fall migratory period and many of these individuals are likely engaging in migration.”
Line 52: In talking about migratory animals, it is very important to use clear and replicated terminology throughout. For example, here, the use of “local populations” is not clear. In the context of this paper, “local” is referring to animals at their summer site of residency (e.g. where they grew their fur). But they could be called “local” to other places at other times of the year. Include an explicit definition and use throughout.
Line 56: Remove “or are”
Line 60: “goals” should be plural here, I think.
Line 75: This seems like an improper use of “relative”. Reword?
Line 80: What is this list of references demonstrating? Just paper that have looked at “migratory tree bats” using stable isotopes (in which case Fraser et al. 2012 should be replaced with Fraser et al. 2017) or all papers looking at bat migration using stable isotopes (in which case many references are missing – much of the work from Dr. Voigt’s lab, for example)?
Line 103: Similar to my comment about line 52 – clarity is needed in the wording here. When you are referring to “sourcing” bats, what time frame is being addressed?
Line 152: I am not at all familiar with the details of trace element analysis and found this confusing. If some of the concentrations of some of the trace elements are calculated/inferred based entirely on the concentration of other elements, then what are these secondary (inferred) element results adding to ultimate analysis that isn’t already provided by the primary (measured) ones? I don’t doubt that this method is appropriate, but I would appreciate a little more explanation.
Line 172: I recommend adding a #5 here (or maybe adding to #4) stating that the paper assesses the accuracy and precision of the origin assignments.
Line 169/170, 173: I think it is more appropriate to say that the study is assessing variation “among” individuals as opposed to “in” them, as “in” suggested intra-individual variation.
Line 229: I think this reference should be Fraser et al. 2017
Line 269: Change “between” to “among”
Line 294: Change “is” to “are”.
Line 312: Awkward sentence, reword
Line 322: I think it is inferred, but I recommend explicitly stating that all precision measurements are based on >0.5 probability accuracy threshold.
Line 333: Grammatical error here, reword
Line 354: I recommend stating somewhere in the paper the dates when the bats that were sampled were collected. I think that the readers are referenced to the Pylant paper for this info – it would be more convenient to have the dates in the present paper.
Line 370: Change “use” to “using”
Line 388: Awkward wording, reword
Line 396: Some words missing here.
Line 418: Change “combing” to “Combining”
Table 3: I’m uncertain about what is going on with the samples that had a 0 probability of being from the collection location and 0 precision. The table says they were “incorrect”. Is this experimental error? If these results are real (e.g. trace element results are correct), then what is going on? It might be interesting to mention these two points in the discussion.

Reviewer 3 ·

Basic reporting

This papers deals with the use of trace elements for assigning the likely spatial origin of bats killed at wind turbines in the US. Previously, geographic assignments were mostly conducted via the use of stable hydrogen isotope ratios in fur keratin. Here, the authors used a novel approach by analyzing trace element concentration in bat fur. Overall the paper is well written. It is widely assumed that trace elements show up in consumer tissue without any offset compared to the baseline enrichments of food items. The authors make the point that their methods allowed the assignment of 80% of the carcasses, while reducing likely geographic locations in half.

Experimental design

It would have been instructive to do a spatial analyses based on stable hydrogen isotope ratios in fur keratin to see if the maps look similar, or to assess which of the approaches provide a more fine-scale resolution.

Line 135: the authors should specify from which part of the torso they collected fur. Previous molting studies focused mostly on fur from the interscapular region of the back. If authors varied in the area from which fur was collected, they may vary in the time period of molt as well, since this is specific for specific parts of the body.

Validity of the findings

Line 195: For geographic assignments, the authors created basemaps by extrapolating values. Usually, the method of choice is some kriging approach. they need to specficy this. Importantly, the authors' approch does not take intra-site variation into account. Reference sampling sites are unknown, which could simulate a fine-scale resolution. yet this might be an artefact of ignoring intra-site variability.

Line 244: the authors chose a 50% cut-off point for accepting a geographic assignment as correct. This approach is not very conservative and thus error-prone. A higher cut-off point of around >80% seems to be more desirable from my point of view.

Additional comments

The paper starts with some alarming numbers that in Northamerica about 70 bats are killed per year and wind turbine. This is a very high number indeed and suggests that populations of affected bats will likely collapse at some point soon. The authors need to provide good documentation for these numbers.

Line 43: the authors should specify that they are talking about the Northamerican scenario. Obviously, other species of bats are affected in Europe, Latin America etc. The authors should go through their manuscript and check if they make generic statements for bats, but mean Northamerican bats. They should specify which geographic background they are referring too, since PeerJ is an international journal (not the Midwestern Naturalist) and therefore it is read by an international readership. I suggest to broaden the geographic focus by citing international papers.

In the introduction, the authors should double-check if they considered all publications on bats and trace elements. One paper about mineral-drinking bats from the Amazon comes into my mind (Ghanem et al. 2013 Journal of Tropical Ecology; Hernout et al. 2016 Chemosphere, 2016 Environmental Pollution; Rahman et al. 2020 Environmental Chemistry and Ecotoxicology among others). I am sure that there are others and therefore, it would be good to review the literature on this topic in some additional section of the introduction.

---

## Round 0.2 · accepted · Accept

· Academic Editor

Accept

As you can see, only one reviewer replied and they found your revised version to be adequate for publication. The other reviewers either did not reply or were not able to submit their review on time. From this, I conclude that your work is no ready for publication.

Reviewer 1 ·

Basic reporting

NA

Experimental design

NA

Validity of the findings

NA

Additional comments

The authors did a fantastic job of reconciling all reviewer comments. The manuscript is much improved and should be of interest to those studying bats as well as those interested in the topic of geographic assignment. Well done.